# Knockdown of a Novel Gene *OsTBP2.2* Increases Sensitivity to Drought Stress in Rice

**DOI:** 10.3390/genes11060629

**Published:** 2020-06-08

**Authors:** Yong Zhang, Limei Zhao, Hong Xiao, Jinkiat Chew, Jinxia Xiang, Kaiyun Qian, Xiaorong Fan

**Affiliations:** 1State Key Laboratory of Crop Genetics and Germplasm Enhancement, Nanjing Agricultural University, Nanjing 210095, China; 2017203041@njau.edu.cn (Y.Z.); 2018103101@njau.edu.cn (L.Z.); 2019103098@njau.edu.cn (H.X.); jkchew1986@njau.edu.cn (J.C.); 2019203051@njau.edu.cn (J.X.); qiankaiyun@njau.edu.cn (K.Q.); 2Key Laboratory of Plant Nutrition and Fertilization in Lower-Middle Reaches of the Yangtze River, Nanjing Agricultural University, Nanjing 210095, China

**Keywords:** transcription factor IID, *OsTBP2.2*, drought stress, photosynthesis, rice

## Abstract

Drought stress is a major environmental stress, which adversely affects the biological and molecular processes of plants, thereby impairing their growth and development. In the present study, we found that the expression level of *OsTBP2.2* which encodes for a nucleus-localized protein member belonging to transcription factor IID (TFIID) family, was significantly induced by polyethylene glycol (PEG) treatment. Therefore, knockdown mutants of *OsTBP2.2* gene were generated to investigate the role of OsTBP2.2 in rice response to drought stress. Under the condition of drought stress, the photosynthetic rate, transpiration rate, water use efficiency, and stomatal conductance were significantly reduced in *ostbp2.2* lines compared with wild type, Dongjin (WT-DJ). Furthermore, the RNA-seq results showed that several main pathways involved in “MAPK (mitogen-activated protein kinase) signaling pathway”, “phenylpropanoid biosynthesis”, “defense response” and “ADP (adenosine diphosphate) binding” were altered significantly in *ostbp2.2*. We also found that *OsPIP2;6*, *OsPAO* and *OsRCCR1* genes were down-regulated in *ostbp2.2* compared with WT-DJ, which may be one of the reasons that inhibit photosynthesis. Our findings suggest that *OsTBP2.2* may play a key role in rice growth and the regulation of photosynthesis under drought stress and it may possess high potential usefulness in molecular breeding of drought-tolerant rice.

## 1. Introduction

Abiotic stress has a significant inhibitory effect on plant growth and development, and even affects crop productivity in severe cases [1,2,3,4,5,6]. Among which, drought stress is the most prevalent environmental threat in the agricultural sector, affecting approximately 75% of the global harvested areas and leading to a significant decline in crop yield worldwide [7]. Water deficit caused by drought stress has been reported to interfere with many aspects of the normal plant processes such as plant cellular metabolism, biochemical processes and physiological functions. Photosynthesis is one of the essential plant biological processes which is particularly sensitive to water deficit. During drought stress, photosynthetic metabolism was severely inhibited, including photophosphorylation [8], ATP (adenosine triphosphate) synthesis [9], and subsequently lead to significant decline in the photosynthetic rate [10]. Studies have shown that chloroplasts, the primary site for photosynthesis, are highly sensitive to dehydration; they will affect the metabolism of carbon and energy, which will affect the growth of plants under dehydration [11]. In the absence of water, stomata are closed to reduce water loss, but it also leads to a reduction in CO_2_ supply and thus reduces the consumption of ATP and NADPH (nicotinamide adenine dinucleotide phosphate) in the thylakoid [12,13].

In order to cope with drought stress, plants have evolved various strategies at physiological, biochemical and molecular levels to enhance their adaptability to drought stress and increase the chance of survival. The abscisic acid (ABA) pathway is an important strategy for regulating plants’ response to drought stress and optimizing water use efficiency [14]. Drought stress will stimulate the production and accumulation of ABA in different organs of the plants and can activate the transmission of downstream signals, thereby improving the drought resistance of the plants [15,16]. At the molecular level, a significant number of transcription factors have been reported to be involved in the modulation of plant response to drought stress. In Arabidopsis, *MYB96* transcription factor mediates TAG (triacylglycerol )biosynthesis, which not only triggers plant adaptive responses, but also optimizes energy metabolism to ensure plant adaptation under stress conditions [17]. Under drought conditions, the stress-induced apetal2/ethylene responsive factor (AP2/ERF) transcription factor *AtTINY* regulates plant responses to drought by activating related gene transcription and abscisic acid-mediated stomatal closure [18]. Knockout of the zinc finger transcription factor gene *BBX24* reduced the tolerance of Chrysanthemum (*Chrysanthemum morifolium*) to freezing and drought stress [19]. In addition, drought stress can trigger rapid accumulation of some signaling molecules such as abscisic acid (ABA) to modulate plant response to water deficit [20,21]. Recent studies have shown that *OsGHR1* (Guard cell Hydrogen peroxide-Resistant1) can participate in the early transduction pathway of ABA signaling [22].

TFIID is composed of TATA-box binding protein (TBP) and TBP-associated factor (TAFs), which plays a key role in the preinitiation of RNA polymerase II transcription [23,24]. TATA binding protein can bind to a particular DNA sequence, known as TATA box. Studies have shown that genes containing TATA box sequence in promoter region usually responds to different environment stresses [25]. The transcription factors that make up TFIID under abiotic stress can enhance plant’s resistance and adaptability to the stress environment [26]. In *Arabidopsis thaliana*, some genes belonging to the TFIID family have been shown to play an active role in dealing with abiotic stress conditions, such as *AtTAF10* (TBP-associated factor 10) [27]. Finger millet transcriptome analysis, an important C4 species, revealed that TATA binding protein (TBP) associated factors (TAFs) and other basic regulatory genes have strong reactivity to drought stress [28]. Moreover, it has also been reported that the expression level of *EcTAF6* encoding for a TATA-box binding protein associated factor 6, increased significantly under abiotic stresses such as salt, osmotic and oxidative stresses. Therefore, the gene *EcTAF6* may play a crucial role in the protection and stress-related transcription process [29].

Rice is the major staple food which feeds more than 50% of the global population. However, as the main crop which lives mainly in flooded conditions, the semiaquatic nature of rice makes it more sensitive to drought stress compared with other cereal crops such as barley (*Hordeum vulgare*), rye (*Secale cereale*) and wheat (*Triticum aestivum*). Therefore, it is important to study the mechanism of the drought response in rice for the development of drought-tolerant rice variety. However, little is known about the function of TATA-box binding protein in the drought stress response in rice. In the present study, we tested the function of a putative TFIID binding protein OsTBP2.2 in the response of rice to the drought condition by mimicking drought stress using PEG4000 and PEG6000 hydroponically. Subcellular localization analysis revealed that OsTBP2.2 is localized in the nucleus. Knockdown mutant of *OsTBP2.2* exhibited significantly reduced photosynthetic rate under drought condition, compared with wild type (WT-DJ). Transcriptomic analysis of *OsTBP2.2* knockdown mutant revealed that *OsTBP2.2* may involve in the regulation of the processes in photosynthesis under drought stress. Collectively, these results suggest that *OsTBP2.2* may play an important role in drought stress response and tolerance in rice.

## 2. Materials and Methods

### 2.1. Subcellular Localization Analysis

The full-length open reading frame (ORF) of *OsTBP2.2* (Os10g0432300) was amplified from *Oryza sativa*. The vectors constructed for subcellular analysis were designated as p35S:OsTBP2.2:eGFP and p35S:eGFP:OsTBP2.2. The vectors were transformed into *Agrobacterium tumefaciens* (EHA105). *A. tumefaciens* strain EHA105 harboring the vectors was then infiltrated into the leaves of *Nicotiana benthamiana*. After 72 h, the green fluorescent protein (GFP) fluorescence was observed and photographed by confocal laser scanning microscope (Leica SP8).

### 2.2. Plant Materials and Growth Conditions

In this study, the *O. sativa* japonica cultivar “Dongjin” was used as wild type (WT-DJ), because Dongjin is the cultivar in which the selected mutants were generated. The *ostbp2.2* mutant lines (3A-07008, 3A-11562) were obtained from the rice genome database (http://signal.salk.edu/cgi-bin/RiceGE). The DNA samples were extracted using the SDS (Sodium dodecyl sulfate) method and used for genotyping PCR. The primers are listed in Appendix A.

### 2.3. Drought Stress Treatment

For hydroponic drought stress experiment, the seeds were surface-sterilized and germinated in the IRRI solution. IRRI (International Rice Research Institute) solution contained 0.35 mM K_2_SO_4_, 0.3 mM KH_2_PO_4_, 1 mM MgSO_4_, 1 mM CaCl_2_, 0.5 mM Na_2_SiO_3_, 20 µM H_3_BO_3_, 9 µM MnCl_2_, 20 µM EDTA-Fe, 0.77 µM ZnSO_4_, 0.32 µM CuSO_4_, and 0.39 µM (NH_4_)_6_Mo_7_O_24,_ pH 5.5. The seedlings which were used in this study were treatment with 15% (*w*/*v*) PEG4000 and PEG6000 for one week in a growth room, 14 h light (30 °C)/10 h dark (22 °C) photoperiod and 60% relative humidity [6,30], and the light intensity was about 1500 μmol m^−2^ s^−1^. After one week of the drought stress experiments, the photosynthesis rate, chlorophyll content and water use efficiency were measured.

For the limited water supply experiment, wild type and mutant lines were cultivated in the plot with the same soil conditions for two weeks under normal water supply. Then, water supply was withheld for one week to maintain a dry state and re-supplied after one week. Chemical properties of the soils included total nitrogen content, 0.6 g/kg; available P content, 18.9 mg/kg; exchangeable K, 185.7 mg/kg and pH 6.5. The height and the maximum diameter of the pot was 18 cm and 20 cm, respectively. Each pot was filled with about 4 kg of soil and planted with 10 equal-sized rice seedlings. Rice seedlings were then cultivated in a greenhouse with a 12 h light (30 °C)/12 h dark (22 °C) photoperiod. The relative humidity and the maximum light intensity were controlled at about 60% and 1500 μmol m^−2^ s^−1^, respectively.

### 2.4. RNA Extraction and qRT-PCR Assay

The samples were taken from the shoot and root tissues, each line was taken three biological replicates, then RNA was extracted from the samples using TRIzol reagent. qRT-PCR was performed using Applied Biosystems StepOnePlus Real-Time PCR System for the expression of the genes, *OsActin* was selected as the internal standard. The primers are listed in Appendix A.

### 2.5. Measurement of Photosynthetic Characteristics

In brief, the photosynthetic parameters, including transpiration rate, photosynthetic rate, stomatal conductance and related data, the fully expanded leaves of rice seedlings were measured between 9:00 am and 11:00 am using Li-COR6400 portable photosynthesis system (Li-COR, Lincoln, NE, USA) with LED leaf cuvette. When measuring, the photosynthetically active radiation, leaf temperature and CO_2_ concentration were set to 1500 μmol m^−2^ s^−1^, 25 °C and 400 ppm. Five biological replications for each line under control and PEG treatment.

### 2.6. RNA-seq Analysis

RNA samples were isolated from WT-DJ and the *ostbp2.2* line (3A-11562) for RNA-seq analysis, with three biological replicates per genotype. The filtered reads were aligned with the reference genomes (Nipponbare) using HISAT2 software (open-source software freely available at https://daehwankimlab.github.io/hisat2/). We used HTSeq to align the read count value on each gene as the original expression level of the gene, and then normalized the expression level using fragments per kilo bases per million fragments (FPKM) [31]. DESeq software was used for identifying genes differentially expressed between WT-DJ and 3A-11562. Differentially expressed genes (DEGs) were defined as genes having the expression difference of |log_2_FoldChange| > 1 and a *p*-value < 0.05 [32,33]. The identified DEGs were analyzed by KEGG (kyoto encyclopedia of genes and genomes)pathway and Gene Ontology (GO) term enrichment [34,35].

### 2.7. Statistical Analysis of the Data

All data were analyzed by ANOVA using the statistical SPSS 11.0 statistical software (SPSS Inc., Chicago, IL, USA). The different letters indicate a significant difference between the mutant lines and WT-DJ (*p*< 0.05, one-way ANOVA).

## 3. Results

### 3.1. Expression Pattern and Subcellular Localization of OsTBP2.2

In order to determine the spatial expression pattern of *OsTBP2.2* gene in rice, we performed qRT-PCR analysis towards the RNA samples isolated from leaf blade, culm, and panicle. The results showed that *OsTBP2.2* gene was constitutively expressed in all organs tested, however its expression level was significantly higher in leaf blade than that of culm and panicle, which were 87% and 71% higher, respectively (Figure 1). Meanwhile, the expression level of *OsTBP2.2* in leaf blade II was approximately 18% higher than that of leaf blade I (Figure 1). The expression of *OsTBP2.2* in culms and panicle was not significantly different (Figure 1).

To determine the subcellular localization of OsTBP2.2, green fluorescent protein (GFP) reporter gene was fused to the open reading frame sequence of *OsTBP2.2*, with expression driven by CaMV 35S promoter. Vector carrying GFP gene under the control of CaMV 35S served as positive control. Then, infection solution of Agrobacterium harboring the resulting plasmids was injected into the epidermis of the tobacco leaves. The data showed that the p35S::TBP2.2::GFP and p35S::GFP::TBP2.2 fusion protein were confined to the nucleus (Figure 2).

### 3.2. Response of OsTBP2.2 Gene Expression to Drought Stress

Previous studies showed that polyethylene glycol (PEG) can be used to simulate drought stress without causing toxic effect on the plant, and PEG6000 has been used frequently in many studies on drought stress response in plant [36,37,38]. Therefore, we treated the wild type plant (WT-DJ) with PEG6000 solution with the concentration of 15% to analyze the expression pattern of *OsTBP2.2* in response to drought stress using relative quantitative PCR. The data showed that the expression of *OsTBP2.2* was significantly elevated at 3 h after 15% PEG6000 treatment, decreased slightly at 12 h, and again increased at 24 h (Figure 3). It indicated that *OsTBP2.2* may play an important role in rice response to drought condition.

### 3.3. Characterization of T-DNA Insertion Mutants, ostbp2.2

To determine the function of *OsTBP2.2* in rice, we obtained two independent *OsTBP2.2* T-DNA insertion mutants which were designated as 3A-07008 and 3A-11562 (Appendix A). In the line of 3A-07008, the 3′UTR of *OsTBP2.2* was inserted with T-DNA; and in the line of 3A-11562, the promoter of *OsTBP2.2* was inserted with T-DNA (Appendix A). Relative quantitative PCR analysis showed that the expression of *OsTBP2.2* gene was significantly reduced in the leaf blade, culms and panicles of *ostbp2.2* compared with that of WT-DJ (Figure 4D). Phenotypically, *ostbp2.2* had shorter plant height and fewer tillers compared with the wild-type (WT-DJ) (Figure 4A–C). In addition, the dry biomass of the leaf blades, culms, sheath and panicles of *ostbp2.2* were significantly lower than that of WT-DJ, at both the flowering stage and the mature stage (Appendix A).

### 3.4. Knockdown of OsTBP2.2 Affect Rice Growth and Photosynthesis at Seedling Stage under PEG Treatment

To learn more about the role of *OsTBP2.2* in response to drought stress toward rice, the *ostbp2.2* lines and non-transgenic WT-DJ were grown in three different nutrient solutions which are normal nutrient solution, nutrient solution supplemented with PEG4000 and PEG6000 (Figure 5A–C). We observed that the plant height, root length, shoot and root biomass of *ostbp2.2* were lower compared with WT-DJ under all growth conditions (Appendix A). However, the decline rate of plant height of *ostbp2.2* were significantly higher under PEG4000 and PEG6000 solution treatment compared with WT-DJ, increased by approximately 8% under PEG4000 treatment and 10% under PEG6000 treatment for plant height (Figure 5D). The decline rate of root length was no difference with WT-DJ under PEG4000 and PEG6000 solution treatment (Figure 5E). The decline rate of biomass were significantly increased under both solution treatment compared with WT-DJ, increased by approximately 10% under PEG4000 and 12% under PEG6000 about shoot biomass, increased by approximately 44% under PEG4000 and 23% under PEG6000 root biomass (Figure 5F,G). The result indicated that knockdown of *OsTBP2.2* reduced the drought tolerance of rice.

Compared with WT-DJ, the chlorophyll content was significantly reduced in *ostbp2.2* (Appendix A), however, the decline rate of chlorophyll content was increased by 68% under PEG4000 and by 50% under PEG6000 (Figure 6A). There was no difference in water use efficiency, transcription rate, conductance to H_2_O and photosynthesis rate between WT-DJ and *ostbp2.2* under normal solution (Figure 6B–E). However, under PEG treatment as compared with WT-DJ, the water use efficiency was reduced by 52% under PEG4000 treatment and by 40% under PEG6000 (Figure 6B). Under PEG treatment, the transpiration rate of *ostbp2.2* decreased by approximately 72% in PEG4000 and 46% in PEG6000, compared with that of WT-DJ (Figure 6C). Compared with WT-DJ, the stomatal conductance was reduced by about 75% under PEG4000 and about 60% under PEG6000 in the *ostbp2.2* (Figure 6D). Under PEG4000 and PEG6000 conditions, the photosynthesis rate of *ostbp2.2* lines were 73% and 56% lower than that of WT-DJ, respectively (Figure 6E). These results showed that the negative effect of PEG-mimic drought conditions on the photosynthesis rate of *ostbp2.2* was greater than that of wild-type (Figure 6E).

### 3.5. Knockdown OsTBP2.2 Alter the Resistance to Drought in Soil

As shown above, the growth and development of *ostbp2.2* were inhibited by PEG. Thus, in order to further confirm that OsTBP2.2 is involved in the drought resistance of rice, soil grown WT-DJ and *ostbp2.2* seedlings were exposed to drought stress by withholding the water supply (Figure 7A). After 10 days of drought stress, the *ostbp2.2* lines, 3A-07008 and 3A-11562, shown typical severe dehydration and obvious symptoms of wilting relative to WT-DJ (Figure 7B). The survival rate of seedlings was calculated 7 days after re-watering (Figure 7C). The result shown that 52% WT-DJ seedlings survived, but the survival rate of 3A-07008 and 3A-11562 was only 34% and 30%, respectively (Figure 7D).

### 3.6. Knockdown of OsTBP2.2 Gene Affect Transcriptional Responses of Rice under PEG Treatment

In order to understand the effect of *OsTBP2.2* knockdown on rice transcriptional regulation under drought stress, RNA-seq analysis was conducted on WT-DJ and *ostbp2.2* which was grown in nutrient solution supplemented with 15% PEG6000. Under the criterion of a |log_2_FoldChange| > 1 and a *p*-value < 0.05, 283 down-regulated genes and 637 up-regulated DEGs were identified in *ostbp2* (Figure 8A). GO enrichment analyses shown that the biological processes (BP) affected by the knockdown of *OsTBP2.2* were “protein phosphorylation”, “cell surface receptor signaling pathway” and “response to iron ion”, “signal transduction” and “defense response”. These terms play an important role in regulating the self-regulation of rice during stress [39,40,41,42]. As for the molecular functions (MF) terms, the major subcategories were “ADP binding”, “adenyl ribonucleotide binding” and “adenyl nucleotide binding”. The alteration of these aspects may affect the energy supply which needed during the development of rice, thereby inhibiting growth, such as photosynthesis. Finally, among the cell components (CC), “cell periphery” was the major subcategories, and “plasma membrane” and “integral component of membrane” were also high. These results are shown in Figure 8B.

Kyoto Encyclopedia of Genes of Genomes (KEGG) enrichment analysis of the identified DEGs revealed that “MAPK signaling pathway” and “phenylpropanoid biosynthesis” were enriched (Figure 8C). Currently, the study shown that MAPK signaling pathway link with ABA regulated abiotic stress in plant [43,44]. “phenylpropanoid biosynthetic” pathway is one of the ways for plants to produce secondary metabolism, and the secondary metabolites play an important role in the maintenance of response to adversity [45]. Under PEG6000 treatment, the pathways of “carbon fixation in photosynthetic organisms”, “citrate cycle (TCA cycle)”, “plant hormone signal transduction” and “porphyrin and chlorophyll metabolism” were enriched in the *ostbp2.2* lines (Figure 8C). The enrichment of these pathways was affected by drought stress and affects the growth of rice by regulating biological pathways such as photosynthesis.

### 3.7. RT-PCR Verification of DEGs under PEG Treatment

In order to verify the accuracy of RNA-seq data, we selected 8 genes in DEGs for RT-PCR verification. Under normal IRRI solution, except for the down-regulation of *OsPiz-t*, the genes of *OsPIP2;6*, *OsZFP6*, *OsGHR1*, *OsRCCR1*, *OsPAO*, *OsRLCK135* and *OsDREB2a* were not significantly different between *ostbp2.2* lines and WT-DJ, while under PEG6000 treatment, the expression of all the selected genes were significantly down-regulated compared with WT-DJ (Figure 9). These results are consistent with the results of RNA-seq.

## 4. Discussion

Rice as a tropical and subtropical crop is relatively sensitive to drought conditions [46]. Water deficit in agricultural production is one of the main limiting factors for rice growth in many parts of the world [47]. Lack of water resources poses a serious threat to the increase in yields of drought and water-sensitive crops [48,49]. In agricultural production, the prevalence of improved varieties in rice varieties has narrowed their genetic basis, resulting in increased sensitivity of rice to various abiotic stresses [50,51]. Therefore, our findings may provide useful information for molecular breeding programs to develop improved rice cultivars tolerant to drought stress. In rice, the genes related to drought stress response have been found, such as *OsAHL1*, *OsTPP1*, *OsRab7* [52,53,54]. Enhancing the expression of *OsAHL1* can improve drought avoidance and tolerance in rice [52]. The expression of *OsTPP1* was activated under abiotic stress, salt and drought stress [53]. Overexpression of the gene *OsRab7* was improved tolerance to drought stress in rice [54].

The TFIID family genes play a key role in regulating the transcription of other genes, and TATA binding protein is an important component of TFIID. However, there is little research on TATA binding protein in rice. Previous studies have shown that *AtTAF5,* a TBP-associated factor 5, is necessary for the growth of *Arabidopsis* and the mutation of *AtTAF5* will inhibit the inflorescence growth [55]. The *AtTAF6,* a TBP-associated factor 6 mutation specifically affects the growth of pollen tubes, and the transcriptional regulation of a specific base factor set is controlled by this basic transcription factor [56]. The overexpression of *AtTAF10*, a factor of the TFIID complex, enhanced the tolerance to salt stress. Meanwhile, knockdown of the gene increased the sensitivity to salt stress [28]. Although meaningful progress has been made in the study of the biological function of TFIID in Arabidopsis, the study on the biological function of TFIID gene in rice is still limited. Previous studies have shown that the transcription factors constituting TFIID may act as enhancers and positive regulators in the response of plant root to drought stress [26]. In addition, the transcription initiation factor TFIID can regulate gene expression and thus the plants can adapt to osmotic stress [55,56]. In this study, the biological function of the *OsTBP2.2* is exhibited. Our data show that the knockdown of *OsTBP2.2* can affect the growth and development in rice (Figure 4), and *OsTBP2.2* gene has a certain protective effect on rice during drought.

### 4.1. OsTBP2.2 May Affect the Drought Resistance of Rice by Regulating the Transcription of Other Genes

In our results, drought stress can induce transcription of *OsTBP2.2* (Figure 3), implying that *OsTBP2.2* may be regulated by drought stress and affected plant growth. Therefore, T-DNA insertion mutants, *ostbp2.2*, were used to verify the function of *OsTBP2.2* against drought stress. Our results proved that the knockdown of *OsTBP2.2* reduces the rice resistance to drought stress, inhibits the growth of rice and reduced the biomass (Figure 5, Appendix A). Research evidence proved that *AtTAF10*, which constitutes the TFIID complex with TATA binding protein, can be regulated by abiotic stress in Arabidopsis, and the knockdown of *AtTAF10* is sensitive to salt stress [28]. Under abiotic stresses such as drought stress, some physiological processes will undergo adaptive adjustments, sometimes inhibiting the development of some physiological metabolism, so that plants can survive under abiotic stress [57,58,59]. The knockdown of *OsTBP2.2* alter some GO terms such as “cell surface receptor signaling pathway”, “signal transduction” and “defense responses” which is related to resistance (Figure 8B). Furthermore, the knockdown of *OsTBP2.2* affect the pathways such as “MAPK signaling pathway-plant” and “phenylpropanoid biosynthesis” (Figure 8C), which connected with environmental stresses and developmental programs [45,60,61,62,63]. Under abiotic stress, plants generally produce a large amount of reactive oxygen species, which can damage the mitochondria, chloroplasts and cell membranes of plants unless they are removed in time [64,65,66]. Under abiotic stress, the gene *OsZFP6* participates in the function of antioxidant defense mechanism [67]. However, the transcription of *OsZFP6* is suppressed in *ostbp2.2* under drought stress compared with WT-DJ (Figure 9B), which may also lead to the accumulation of reactive oxygen species, leading to oxidative damage to the cells, thereby inhibiting the normal growth of rice. Besides that, when subjected to drought stress, the expression of the genes related to the regulation of drought resistance, such as *OsPiz-t*, *OsRLCK135* and *OsDREB2a*, were also significantly inhibited in *ostbp2.2* (Figure 9C,G,H). These genes play an important role in the resistance of rice to environmental stresses [68,69,70]. This evidence showed that *OsTBP2.2* may affect the rice growth under drought stress by regulating the expression of stress resistance genes.

### 4.2. Regulation of Photosynthesis by OsTBP2.2 under Drought Conditions

Under drought stress, the knockdown of *OsTBP2.2* resulted into higher decline in the transpiration rate, stomatal conductance and photosynthetic rate of rice seedlings as compared with WT-DJ (Figure 6C–E). Photosynthesis is one of the primary physiological pathway which is severely affected by drought condition. Studies have reported that the knockdown of *OsGHR1* will inhibit the activation of calcium channel and abscisic acid (ABA) and H_2_O_2_, resulting in stomatal closure [22]. The closure of stomata will limit the rate of CO_2_ exchange between leaf cell and the atmosphere, and ultimately lead to a reduction in photosynthesis efficiency. In the present study, the expression of *OsGHR1* was significantly reduced in *ostbp2.2* under drought stress (Figure 9D), which may also be the reason for the inhibition of *ostbp2.2* photosynthetic rate. Studies have shown that *OsPAO* and *OsRCCR1* are strongly involved in chlorophyll degradation and silencing of *OsPAO* and *OsRCCR1* by RNA inference (RNAi) led to degradation of chlorophyll in rice leaves [71]. At the same time, we detected the expression of *OsPAO* and *OsRCCR1* in *ostbp2.2* lines, the expression of *OsPAO* and *OsRCCR1* were significantly down-regulated under drought stress (Figure 9E,F). It indicated that *OsTBP2.2* perhaps regulated the transcription of *OsPAO* and *OsRCCR1*, thereby regulated the reaction efficiency of photosynthesis in rice chloroplasts.

In the lines of *ostbp2.2*, the expression of *OsPIP2;6*, an aquaporin gene which is specifically expressed in rice leaves, was decreased under drought stress (Figure 9A). Studies have shown that aquaporins play a key role in maintaining the water balance and cell osmotic pressure among various tissues in many plants under drought condition [72,73,74,75]. In order to reduce water loss in plants under drought stress, the plant will close or reduce the opening of stomata, while reducing water loss, it also reduces the entry of carbon dioxide, resulting in weakened photosynthesis [76]. In the plant thylakoid, water is oxidized to molecular oxygen in response to light, the activity of aquaporin can promote the rapid response of thylakoid cavity to light signal [77]. Study expounded that drought stress can induce the up-regulation of *OsPIP2;6* and enhance the water transport in rice [78], and the overexpression of *OsPIP2;6* can increase rice growth rate [79]. However, knockdown of *OsTBP2.2* inhibited the expression of *OsPIP2;6* under drought stress (Figure 9A), and this may make the water entering the cell unable to meet the needs of photosynthesis.

ADP-binding proteins are involved in the conversion of ADP to ATP, and low ATP content will inhibit the biosynthesis of ribulose biphosphate (RuBP) and may limit the photosynthetic assimilation of CO_2_ under drought stress [12,80]. Furthermore, through GO analysis, knockdown of the *OsTBP2.2* affected “ADP binding”, “adenosine ribonucleotide binding” and “adenylate nucleotide binding”, thus affecting the demand for ATP during rice photosynthesis (Figure 8B). The results suggest that *OsTBP2.2* may regulate photosynthesis under drought stress by affecting the expression of aquaporin gene and molecular pathways related to photosynthesis.

In summary, the gene *OsTBP2.2* may exert a certain protective effect on the biological function in response to drought stress in rice. In regard to knockdown *OsTBP2.2* in rice, the growth inhibition ability was enhanced and the photosynthesis was reduced under drought stress. Increasing the expression of *OsTBP2.2* may enhance resistance to drought stress in rice, while *OsTBP2.2* gene may have natural allelic variation in rice, which may be valuable for breeding drought resistant varieties.

## 5. Conclusions

Conclusively, the present study showed that knockdown the gene ***OsTBP2.2*** not only inhibited the growth of rice but also increased the sensitivity to drought stress. According to RNAseq data analysis, it is possible that knockdown ***OsTBP2.2*** altered the pathways of “MAPK signaling pathway”, “phenylpropanoid biosynthesis”, “defense response”, “ADP binding” and inhibited the transcription of ***OsPIP2;6***, ***OsPAO*** and ***OsRCCR1***. So that under drought stress, knockdown *OsTBP2.2* reduced the photosynthesis rate, transpiration rate, water use efficiency and stomatal conductance of rice. 

## Figures and Tables

**Figure 1 genes-11-00629-f001:**
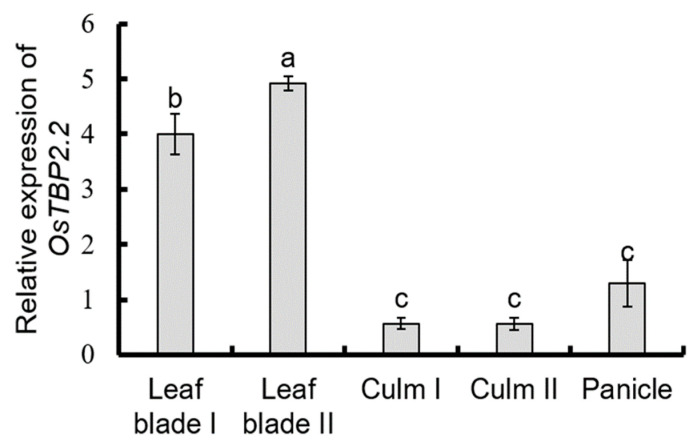
*OsTBP2.2* expression pattern in different parts of rice. Rice RNA samples were collected during the flowering period and the expression of the gene OsTBP2.2 was detected in leaf blade I, leaf blade II, culm I, culm II and panicle. Error bars: SE (*n* = 3). Significant differences are indicated by different letters.

**Figure 2 genes-11-00629-f002:**
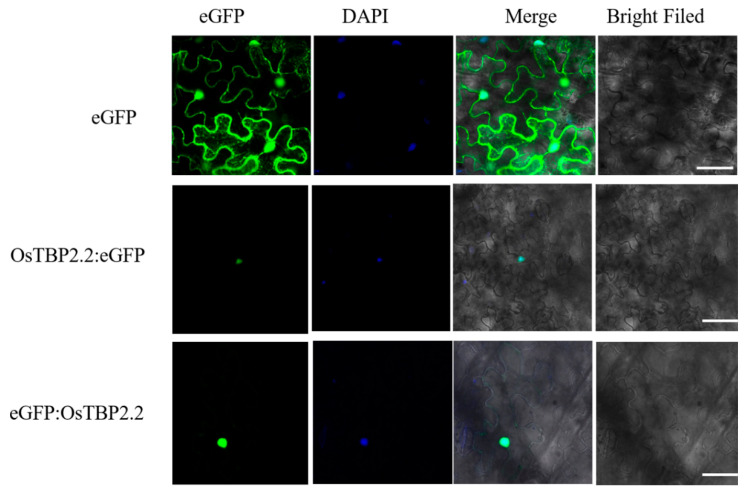
Subcellular localization of OsTBP2.2 Tobacco (*N. benthamiana*) leaves were transiently transformed by *Agrobacterium tumefaciens* harboring plasmid expressing OsTBP2.2::GFP and GFP::OsTBP2.2 fusion protein. Images were taken under a confocal laser scanning microscope in dark field for green fluorescence. Dark cultivation for three days in a growth room, 14 h 30 °C/10 h 22 °C and 60% relative humidity. DAPI (4’6-diamidino-2-phenylindole) staining was used to mark the nucleus. Bars = 5 μm.

**Figure 3 genes-11-00629-f003:**
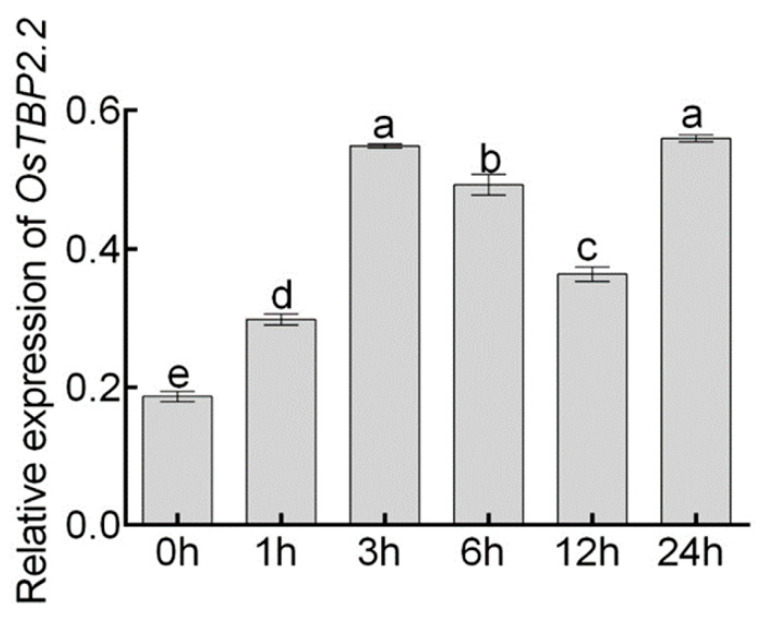
Relative expression analysis of *OsTBP2.2* under PEG treatment. Relative expression level of *OsTBP2.2* in the shoot of WT-DJ under drought stress. Rice seedlings were treated with IRRI solution containing 15% PEG6000 and harvested at different time points (0, 1, 3, 6, 12 and 24 h). Error bars: SE (*n* = 3). Significant differences are indicated by different letters. (*p* < 0.05).

**Figure 4 genes-11-00629-f004:**
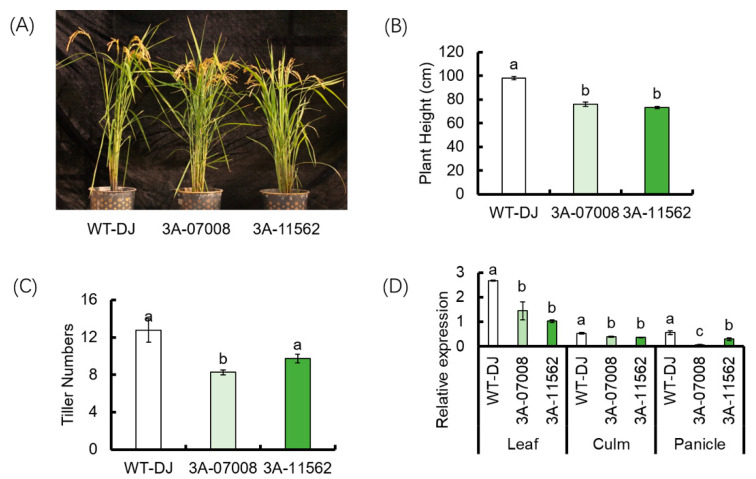
Characterization of the *OsTBP2.2* mutant lines. (**A**) Phenotype of T-DNA insertion mutants of *OsTBP2.2* (3A-07008 and 3A-11562) and the wild type (WT-DJ) under non-stress conditions. (**B**) Plant height of *ostbp2.2* (3A-07008 and 3A-11562) and WT-DJ. (**C**) Tiller numbers of *ostbp2.2* (3A-07008 and 3A-11562) and WT-DJ. (**D**) Real-time quantitative PCR analysis the expression of *OsTBP2.2* in the leaf, culm and panicle of *ostbp2.2* and WT-DJ. Error bars: SE (*n* = 3). Significant differences between *ostbp2.2* and WT-DJ are indicated by different letters. (*p* < 0.05).

**Figure 5 genes-11-00629-f005:**
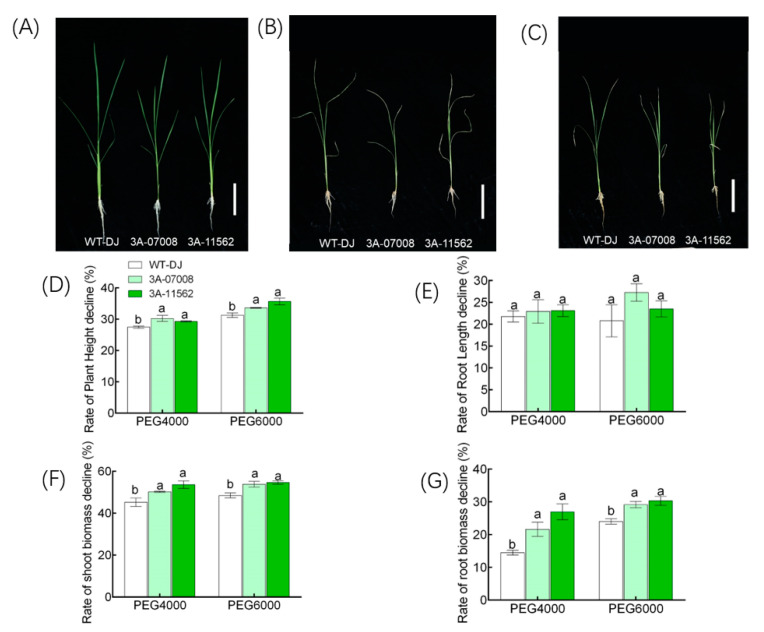
Phenotype of *ostbp2.2* and WT-DJ at seedling stage under control and PEG treatment. Growth performance of *ostbp2.2* and WT-DJ under control (**A**), 15% PEG4000 (**B**) and 15% PEG6000 (**C**). (**D**) Rate of plant height decline in *ostbp2.2* and WT-DJ between control and PEG treatment (PEG4000 and PEG6000 treatment). (**E**) Rate of root length decline in *ostbp2.2* and WT-DJ between control and PEG treatment. (**F**,**G**) Rate of dry weight decline in *ostbp2.2* and WT-DJ between control and PEG treatment, shoot (**F**) and root (**G**). Error bars: SE (*n* > 3). Significant differences between *ostbp2.2* and WT-DJ are indicated by different letters. (*p* < 0.05).

**Figure 6 genes-11-00629-f006:**
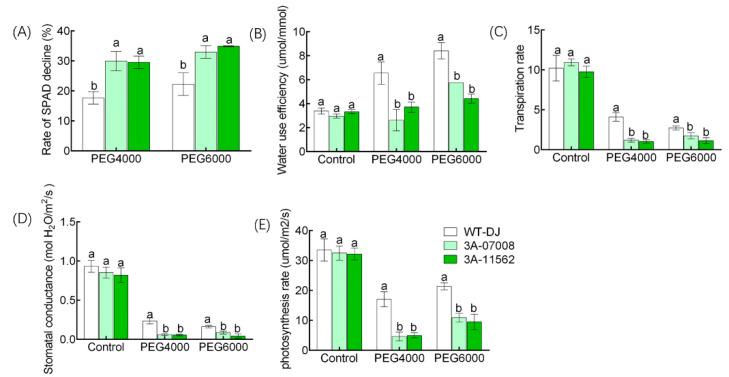
Physiological parameters of *ostbp2.2* and WT-DJ. Four-week-old rice of *ostbp2.2* and WT-DJ were cultivated one week under normal solution, 15% PEG4000 and 15% PEG6000. Rate of SPAD decline (**A**), water use efficiency (**B**), transpiration (**C**), stomatal conductance (**D**) and photosynthesis rate (**E**) were assayed. SPAD, the relative content of chlorophyll. Error bars: SE (*n* = 5). Significant differences between *ostbp2.2* and WT-DJ are indicated by different letters. (*p* < 0.05).

**Figure 7 genes-11-00629-f007:**
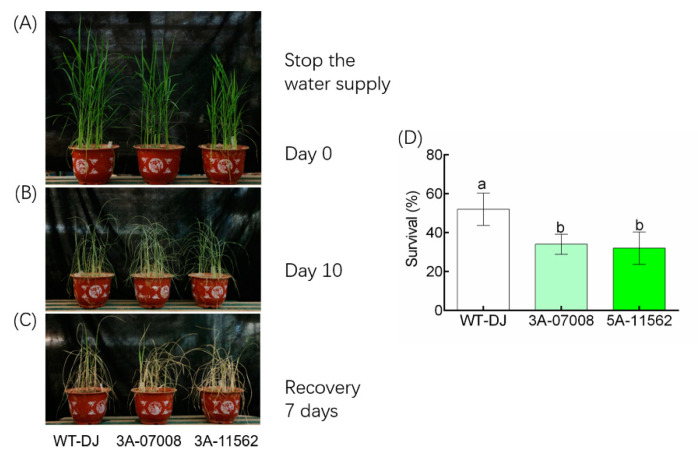
Drought stress sensitivity of *OsTBP2.2* transgenic lines at the seedling stage. (**A**) The seedlings of *ostbp2.2* and WT-DJ were cultivated for 3 weeks by full irrigation in soil, then water supply was withheld for ten days. (**B**) Phenotype of stop water supply ten days. (**C**) Phenotype of the seedlings one week after re-watering. (**D**) Seedling survival of *ostbp2.2* and WT-DJ after re-watering. Error bars: SE (*n* = 5). Significant differences between *ostbp2.2* and WT-DJ are indicated by different letters. (*p* < 0.05).

**Figure 8 genes-11-00629-f008:**
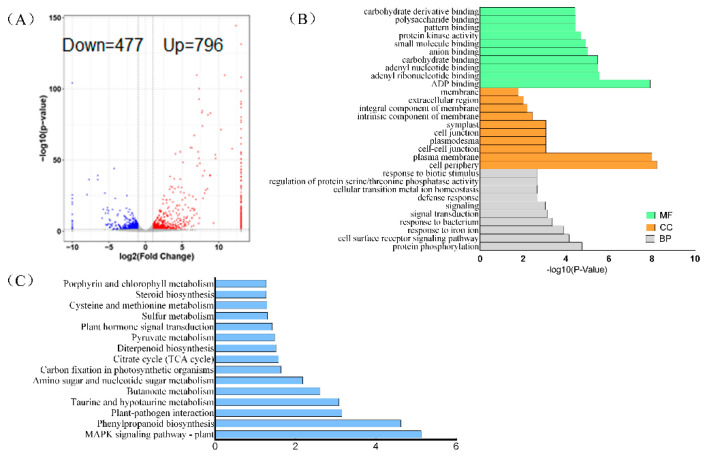
RNA-Seq analysis *ostbp2.2* and WT-DJ under PEG treatment. (**A**) The number of DEGs identified by the RNA-seq results. (**B**) Functional GO analysis of DEGs. According to the results of GO enrichment analysis of differentially expressed genes (DEGs), GO classification is performed according to molecular function (MF), biological process (BP) and cell component (CC), and the top 10 GOs are selected to be the most significant enrichment term entries for display. The standard of significant enrichment is *p*-value < 0.05. (**C**) Results of KEGG enrichment analysis of DEGs.

**Figure 9 genes-11-00629-f009:**
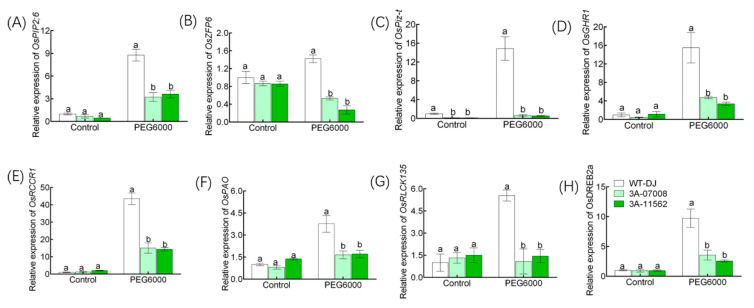
Verification for expressions of eight selected DEGs. Relative expression of the genes in the leaf of *ostbp2.2* and WT-DJ were tested by qRT-PCR after PEG treatment. The tested genes include *OsPIP2;6* (**A**), *OsZFP6* (**B**), *OsPiz-t* (**C**), *OsGHR1* (**D**), *OsRCCR1* (**E**), *OsPAO* (**F**), *OsRLCK135* (**G**), and *OsDREB2a* (H). Control, the seedlings were grown in normal IRRI solution; PEG6000, the seedlings were grown in normal IRRI solution with 15% PEG6000. Error bars: SE (*n* = 3). Significant differences between *ostbp2.2* and WT-DJ are indicated by different letters. (*p* < 0.05).

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
