# Peer review of "Knockdown of a Novel Gene OsTBP2.2 Increases Sensitivity to Drought Stress in Rice"

_genes, 2020, doi:10.3390/genes11060629_

Round 1
Reviewer 1 Report
MS titled "A novel gene OsTBP2.2 for resistance to drought stress in rice" by Zhang et al. provides interesting information on the role of OsTBP2.2. in rice under drought stress. The authors used knockdown mutants of OsTBP2.2 gene. It would be useful if overexpressor of the same gene is available. It would confirm additionally the roles of OsTBP2.2 in drought tolerance. Can authors elaborate on this possibility to include in MS or at least to discuss it.
I pointed out some suggestions and critics directly in pdf file (in att.). Language needs to be improved, particularly part about transcripts analysis.

Author Response
First of all, thank you very much for the valuable comments and suggestions put forward in this article. The questions you have extracted have been modified in the article and some questions have been answered.
Q1. Did you run qPCR of controls (non-treated plants) through all time points?
Initially, during the functional study of the gene OsTBP2.2, we cultured wild-type (WT-DJ) in IRRI solution for 0h, 6h, 12h, and 18h, then detected the expression of the gene OsTBP2.2 by qPCR. The result shown that there is no difference.
The data and the revised article will be displayed in the attached file.

Reviewer 2 Report
The study picks a candidate gene for drought resistance in rice and studies what are the impacts in two mutants selected to have lower expression. These are studied largely in a hydroponic system that uses PEG to simulate drought, and measures growth, photosynthetic parameters and genes expression (both transcriptomics and RTPCR). It’s a useful study. Any study on mutants that sheds light on what any gene does is useful, because slowly we will learn what they all do this way. I have some problems with the study though.
First, I would question the logic of the title. They have shown using mutants that knocking this gene down increases sensitivity to drought. That is not the same as showing this is a gene for drought resistance. They do not show any evidence of increased drought resistance.
Second, I have a problem with the justification made to focus on this gene. I am sure the authors think there is a good chance it plays an important role in drought response. But I do not see the evidence. It may be there in the literature, but the author have yet to present it to the reader in a compelling way. It may be in the results, yet I would say the response of the gene to drought (in terms of expression) is quite small and the evidence that knocking it out makes drought sensitivity higher looks reasonably robust but its not a big effect. Does it justify saying it may have a major role? There are so many studies on mutants and transgenics which claim genes have a major role in drought. In fact, they cannot all be as important as the authors claim for their gene. Thus I think the potential importance of this gene needs to be presented more modestly.
Thirdly, in places its not well written. There are many small errors which I guess stem from translation into English (I have highlighted some but certainly not all). More importantly, they do not take enough care to take the reader with them so the order of paragraphs is sometimes odd, sometimes there is too little information about a gene, sometimes too much. I have tried in places to highlight this below. Also, gene abbreviations are used without telling the reader enough about what the gene is to appreciate the biological significance of the study.
Fourthly, I worry about the statistics. They may be OK but they write nothing about them. It needs a section in the methods. What is the test, how are differences between means determined? I have some doubt about the central hypothesis of this paper and the way its tested. The hypothesis is that there is greater impact of PEG on the mutants than the WT. For me the strongest stats test would be a two way ANOVA, with genotype and treatment plus their interaction tested, and I would like to be convinced that the interaction term is significant. The way I think they have done it, where the mean value of the controls is used to calculate the reduction in the treatments (figure 5) is not as powerful or reliable as an ANOVA as the variation in the controls is not employed in the statistical analysis. There is a risk it overstates significance if there is a lot of uncertainly in the mean value of the control (which figure S3 C for example indicates there might be). I do not want to be too restrictive. The bottom line is, however, whatever method they used should be reported, and I would like to know (from a line in text) that a two way anova showed significant genotype by treatment interaction- or I would like to be convinced that a similarly robust statistical approach had been used.
Those are the big points. More minor ones are below in the order they appear in paper.
15 Is it really strongly induced by drought? There are genes with hundred fold induction to drought.
19 What is WT-DJ? Wild type-?? When I get to the methods, I realise DJ means Dongjin. The abstract should not use abbreviations that are not absolutely obvious.
43-52 This section which focuses largely on some molecular responses is, for me, far to selective in describing processes occurring in drought. Better to first say a little about the hormone responses and then the molecular in a more general way by quoting a good quality review of drought responses before selecting some of the many processes just because they are relevant to this study. I know for a fact there is a study that found the TF Hox genes to be the most responsive genes in rice to drought, yet that is not mentioned. It does not necessarily have to be mentioned, but the reader needs to know the authors are being selective.
58-9 it should be “..enhance a plant’s…” or “…enhance plant….”
67 Something is missing here. The statement “we identified a drought responsive gene….” Is not enough to justify this paper. Please detail the evidence, ideally the reference for the study. If it is not published, why is it not included in this manuscript.
- Add “…used as wild type (WT-DJ) because Dongjin is the cultivar in which the selected mutants were generated.”
91 Light intensity (preferably photosynthetically active radiation (PAR)) should be reported. What is the nutrient solution? Yoshida’s?
94 Details of the pot experiment completely insufficient. Give soil medium, pot size, plants per pot, light temperature and humidity as minimum.
109 There needs to be a section on statistical approach used.
126 …approximately 18 HIGHER than that of….
149 I challenge the use of the word “major” in this sentence. This data alone does not suggest this. Hundreds of genes will be upregulated two or three fold by treatment of this kind. They cannot all play a major role in rice response to drought.
Figure 4 D. I am surprised at the expression levels of the mutants. Why are they not knockout mutants?
Figure 4 and S2; please modify the legend to make it clear that this refers to non-stress conditions.
- I remake more or less the same point as I did above. The statistical test that the impact of PEG on the mutants is the central test for this paper. This line indicates its “significantly”. For me it is critical that the full detail so statistics are given. One approach would be a two way ANOVA with factors genotype and treatment. With this, there must be a significant interaction between the factors. I would expect to see at least quoted the p value of that interaction. At the moment, the reader only has the supplementary figure 3 to see all the means and they do look convincingly like higher impact of drought on the mutants, given that the mutants are much lower for all traits than the WT in control. Even if it is statisticlly significant, the effect is small I think. The best way to show it, however, would be to report the full ANOVA table including the contribution to the variance of all factors (genotype, treatment and interaction).
204 I am not sure what the word “induction” means in this sentence.
220 Just a general comment. The use of simple withholding of water is not a good approach to drought experiments when handling genotypes known to differ in size and rate of water use as in this case because the bigger plant will use water faster and become droughted more quickly. Its OK in this experiment because the smaller genotypes (which presumably experiences a less severe water stress) were actually the ones that suffered the most/ But if the same drought (same soil water potential) have been experienced by all genotypes, I would expect the differences to be more spectacular.
Figure 8. GO terms in B and C are hard to read.
252 I do not follow the logic of the sentence starting “Current, the study shows (not shown) that….” The link with ABA and the KEGG terms is not clear to me. If you believe it, you had better explain it better than this.
259 Word “interfering” is not the right word. Regulating?
263 replace “were no significant….” with “were not significantly…”
270 What is IRRI solution?
274 Replace “..is usually more sensitive…” with “…is relatively sensitive….”
276 Remove However
279 Term ….applying valuable genetic basis…. Is not proper English.
282 Start “Previous studies have shown…”
- I think this paragraph would be improved by a sentence reminding the reader what kind of genes TFIID and AtTAFs are and their connection to the previous paragraph.
286 replace “…,on the contrary..” with “…while,….”
291 English is wrong in this sentence. It is not TFIID that “can adapt to osmotic stress” as this sentence suggest, but the plant.
296 Again, I take exception to the words used in this sentence. Just because a gene is upregulated in drought does not prove it may play an important role…. There must be hundreds of transcription factors that are upregulated by drought.
300 Here or earlier the reader needs to be reminded of the exact relationship between OsTBP2.2 and AtTAF10
318 Again the word “proved” is used wrongly here. I would be OK with a sentence that suggests there is evidence of a role in regulating drought response, but I do not like the word proved (even if these authors use it along with “may” which helps to reduce the strength of their statement. I would suggest simply replacing “proved” with “showed”.
322 I can see no relevance of the first sentence of this paragraph (which mentions ROS which were not even measured in this study) with the rest of the paragraph.
328 OsGHR1 is mentions several times in this manuscript. Nowhere is the reader told what the gene is. I think that applies to several other genes mentioned in the study. The reader will enjoy and understand this paper more if these genes were given names, not just abbreviations. Without some explanation of what these genes do, the discussion does not make a lot of sense.
339 Here is a far better example of how to discuss a particular gene. The opening of the paragraph tells the reader what the gene does. Although in this case, I think perhaps there is too much detail on what aquaporins do. In general, though, in this section I would start a paragraph that states that in the current study gene x is affected by the mutant. Then present literature that says what gene x does. Then come back to your data and interpret your result in light of literature.
343 What does it mean that water oxidation “need activated by aquaporin”. I though aquaporins are water channels. This sentence implies they have a role in the biochemistry of water splitting. I do not think that is right.
355 Would be good to end with some comment on how this might be related to improved drought resistance. For example, perhaps increased expression will enhance resistance, so that experiment should be conducted, and there might be natural allelic variation in rice for this gene and that might be explored.
Additional point. Just curious, what happens to the data on gene expression. In addition to the way its used in this paper, it contains useful information on gene response to drought (of the wild type). Is there any intention to publish it? If not, it would be useful to the community if it was submitted to a database so others could use it.
Author Response
First of all, thank you very much for the valuable comments and suggestions put forward in this article. The questions you have extracted have been modified in the article and some questions have been answered.
Q1. First, I would question the logic of the title. They have shown using mutants that knocking this gene down increases sensitivity to drought. That is not the same as showing this is a gene for drought resistance. They do not show any evidence of increased drought resistance.
Response for Q1. After careful consideration, we think that it will be more rigorous to change the title to "Knockdown of a novel gene OsTBP2.2 increases sensitivity to drought stress in rice ".
Q2. Second, I have a problem with the justification made to focus on this gene. I am sure the authors think there is a good chance it plays an important role in drought response. But I do not see the evidence. It may be there in the literature, but the author have yet to present it to the reader in a compelling way. It may be in the results, yet I would say the response of the gene to drought (in terms of expression) is quite small and the evidence that knocking it out makes drought sensitivity higher looks reasonably robust but its not a big effect. Does it justify saying it may have a major role? There are so many studies on mutants and transgenics which claim genes have a major role in drought. In fact, they cannot all be as important as the authors claim for their gene. Thus I think the potential importance of this gene needs to be presented more modestly.
Response for Q2. Previous studies indicated that the protein members in TFIID family play an important role in the transcriptional regulation. In Arabidopsis and other crop species, protein members in TFIID family (for example, AtTAF5, AtTAF10 and EcTAF6) have been shown to be involved in the plant response to stress. However, the study on the biological function of TFIID genes in rice remains limited, not to mention its involvement in rice response to drought stress. In the present study, OsTBP2.2 is a gene which encodes for a nucleus-localized protein member belonging to transcription factor IID family. Therefore, we think that it is interesting to study the function of OsTBP2.2 in rice response to drought stress.
We completely agree with the comment that the potential importance of this gene needs to be presented more modestly. Therefore, we have changed the word “major” into “important” in the sentence of line 178.
Q3. Thirdly, in places its not well written. There are many small errors which I guess stem from translation into English (I have highlighted some but certainly not all). More importantly, they do not take enough care to take the reader with them so the order of paragraphs is sometimes odd, sometimes there is too little information about a gene, sometimes too much. I have tried in places to highlight this below. Also, gene abbreviations are used without telling the reader enough about what the gene is to appreciate the biological significance of the study.
Response for Q3. We have already supplemented the genes mentioned in the article.
Q4. Fourthly, I worry about the statistics. They may be OK but they write nothing about them. It needs a section in the methods. What is the test, how are differences between means determined? I have some doubt about the central hypothesis of this paper and the way its tested. The hypothesis is that there is greater impact of PEG on the mutants than the WT. For me the strongest stats test would be a two way ANOVA, with genotype and treatment plus their interaction tested, and I would like to be convinced that the interaction term is significant. The way I think they have done it, where the mean value of the controls is used to calculate the reduction in the treatments (figure 5) is not as powerful or reliable as an ANOVA as the variation in the controls is not employed in the statistical analysis. There is a risk it overstates significance if there is a lot of uncertainly in the mean value of the control (which figure S3 C for example indicates there might be). I do not want to be too restrictive. The bottom line is, however, whatever method they used should be reported, and I would like to know (from a line in text) that a two way anova showed significant genotype by treatment interaction- or I would like to be convinced that a similarly robust statistical approach had been used.
Response for Q4. The method that we have employed to analyze the manuscript data have been presented in section 2.7 under materials and methods in lines 143-146.
Two-way ANOVA has also been performed to analyze the data between WT-DJ, ostbp2.2 and PEG treatment, control as requested by reviewer. The analysis results are at the word.
Q5. 15 Is it really strongly induced by drought? There are genes with hundred fold induction to drought.
Response for Q5. We fully agree with the reviewer’s comment that the use of the word “strongly” was an overstatement compared to some other drought-inducible genes. Therefore, we have changed the word “strongly induced” to “significantly induced” in the sentence of line 15. However, we reckoned that although OsTBP2.2 was not as strongly induced by drought as some drought inducible gene, it may still play a role in rice response to drought stress.
Q6. 43-52 This section which focuses largely on some molecular responses is, for me, far to selective in describing processes occurring in drought. Better to first say a little about the hormone responses and then the molecular in a more general way by quoting a good quality review of drought responses before selecting some of the many processes just because they are relevant to this study. I know for a fact there is a study that found the TF Hox genes to be the most responsive genes in rice to drought, yet that is not mentioned. It does not necessarily have to be mentioned, but the reader needs to know the authors are being selective.
Response for Q6. In the revised manuscript, we have added some extra information accordingly.
At line 46 of the revised manuscript, we have added the following sentence:
“The abscisic acid (ABA) pathway is an important strategy for regulating plants response to drought stress and optimizing water use efficiency. Drought stress will stimulate the production and accumulation of ABA in different organs of the plants and can activate the transmission of downstream signals, thereby improving the drought resistance of the plants.”
At line 63-65 of the revised manuscript, we have added the following sentence:
“TATA binding protein can bind to a particular DNA sequence, known as TATA box. Studies have shown that genes containing TATA box sequence in promoter region usually responds to different environment stresses.”
At line 70-74 of the revised manuscript, we have added the following sentence:
“Moreover, it has also been reported that the expression level of EcTAF6 encoding for a TATA-box binding protein associated factor 6, increased significantly under abiotic stresses such as salt, osmotic and oxidative stresses. Therefore, the gene EcTAF6 may play a crucial role in the protection and stress-related transcription process”
Q7. 91 Light intensity (preferably photosynthetically active radiation (PAR)) should be reported. What is the nutrient solution? Yoshida’s?
Response for Q7. As requested, we have included the information on the light intensity (which was 1500 μmol m-2 s-1) in the revised manuscript (line 111). The nutrient solution used in this study was IRRI solution and we have also included the full detail of its ingredient in the revised manuscript (line 105-107).
Q8. 94 Details of the pot experiment completely insufficient. Give soil medium, pot size, plants per pot, light temperature and humidity as minimum.
Response for Q8. The details of the pot experiment have been included in the revised manuscript as requested (line 114-120). Chemical properties of the soils included total nitrogen content, 0.6 g/kg; available P content, 18.9 mg/kg; exchangeable K, 185.7 mg/kg and pH 6.5. The height and the maximum diameter of pot was 18 cm and 20 cm, respectively. Each pot was filled with about 4 kg of soil and planted with 10 equal-sized rice seedlings. Rice seedlings were then cultivated in a greenhouse with a 12h light (30â—¦C) /12h dark (22â—¦C) photoperiod. The relative humidity and the maximum light intensity were controlled at about 60% and 1500 μmol m-2 s-1, respectively.
Q9. Figure 4 D. I am surprised at the expression levels of the mutants. Why are they not knockout mutants?
Response for Q8. T-DNA insertion can have multiple effects on the gene transcription, including the cases of knockout (Min et al., 2019; Leung et al., 2018; Li et al., 2019) and knockdown (Sakuraba et al., 2017; Bashir et al., 2013). In this study, the T-DNA sequence that was inserted into the promoter region and 3’UTR of OsTBP2.2 did not completely silence gene expression, so ostbp2.2 are knockdown mutants.
Min, H.J., Cui, L.H., Oh, T.R., Kim, J.H., & Kim, W.T. (2019). OsBZR1 turnover mediated by OsSK22-regulated u-box E3 ligase OsPUB24 in rice BR response. Plant Journal, 99(47).
Leung, K. P., Luo, M., Gao, C., Zeng, Y., & Jiang, L. (2019). Arabidopsis endomembrane protein 12 contributes to the endoplasmic reticulum stress response by regulating K/HDEL receptor trafficking. The Plant Cell, tpc.00913.2018.
Li, Y., Xu, J., Li, G., Wan, S., & Qi, B. (2019). Protein s-acyl transferase 15 is involved in seed triacylglycerol catabolism during early seedling growth in Arabidopsis. Journal of Experimental Botany, 70(19).
Yasuhito, S., Eun-Young, K., Su-Hyun, H., Weilan, P., Gynheung, A., & Daisuke, T., et al. (2017). Rice phytochrome-interacting factor-like1 (OsPIL1) is involved in the promotion of chlorophyll biosynthesis through feed-forward regulatory loops. Journal of Experimental Botany (15), 15.
Bashir, K., Takahashi, R., Akhtar, S., Ishimaru, Y., & Nishizawa, N. K. (2013). The knockdown of osvit2 and mit affects iron localization in rice seed. Rice, 6(1), 31.
